# Tracking Control for Output Probability Density Function of Stochastic Systems Using FPD Method

**DOI:** 10.3390/e25020186

**Published:** 2023-01-17

**Authors:** Yi Yang, Yong Zhang, Yuyang Zhou

**Affiliations:** 1School of Information Engineering, Inner Mongolia University of Science and Technology, Baotou 014010, China; 2School of Computing Engineering and Built Environment, Edinburgh Napier University, Edinburgh EH10 5DT, UK

**Keywords:** tracking control, probability density function, full probability design, B-spline model

## Abstract

Output probability density function (PDF) tracking control of stochastic systems has always been a challenging problem in both theoretical development and engineering practice. Focused on this challenge, this work proposes a novel stochastic control framework so that the output PDF can track a given time-varying PDF. Firstly, the output PDF is characterised by the weight dynamics following the B-spline model approximation. As a result, the PDF tracking problem is transferred to a state tracking problem for weight dynamics. In addition, the model error of the weight dynamics is described by the multiplicative noises to more effectively establish its stochastic dynamics. Moreover, to better reflect the practical applications in the real world, the given tracking target is set to be time-varying rather than static. Thus, an extended fully probabilistic design (FPD) is developed based on the conventional FPD to handle multiplicative noises and to track the time-varying references in a superior way. Finally, the proposed control framework is verified by a numerical example, and a comparison simulation with the linear–quadratic regulator (LQR) method is also included to illustrate the superiority of our proposed framework.

## 1. Introduction

In recent years, there has been a growing interest in stochastic control, as many systems in the real world, such as those in aerospace, chemical, textile machinery and ships, can all be modelled as stochastic dynamic systems [1]. In the study of stochastic system control, for the Gaussian processes, the distribution can be controlled by only controlling its mean and variance [2]. However, many stochastic systems contain non-Gaussian processes, such as the scale distribution of flocculating particles in the paper process [3], the flame shape distribution [4], and the chemical polymerisation reaction molecular weight distribution [5]. For such stochastic systems with non-Gaussian variables, the mean and variance cannot describe the full statistical information, so that traditional methods are no longer productive [6]. In order to solve this problem, Wang proposed a new stochastic control method by introducing the PDF into the control field in 1996 [7,8]. In this method, the B-spline function is applied to model the PDF, which releases the limitation of the Gaussian assumption of the traditional stochastic control methods [9]. Based on that, a series of stochastic control frameworks have been presented in both practical and theoretical disciplines. On the practical side, Li applied the PDF control method to the fibre length distribution in the grinding process to predict the random distribution of the fibre length and achieved a good control effect [3]. Sun applied this method to the flame temperature field, realised the modelling of the flame temperature field and its iterative learning control, and achieved the purpose of improving the control effect [4]. In theoretical terms, a PDF shape control method using the FPK equation for nonlinear stochastic systems with an arbitrary expression was proposed in [10], where the exact stationary solution of the FPK equation was derived and the controller was designed for a non-polynomial nonlinear function. Reference [11] presented the rational square root B-spline model and introduced the concept of the pseudo-weight, which makes the control algorithm almost unconstrained and the system analysis relatively simple. In these studies, most correlation methods ignore the model error of the B-spline function or simply assume the error as steady-state residuals. Thus, the control methods that have been applied in most of these studies are LQR methods. However, for some complicated practical systems, the PDF model error can be involved in a stochastic dynamical manner [12]. Ignoring that would affect the control performance and even lead to divergence. For such complex practical processes, multiplicative noise, which is also known as state-dependent noise, can better characterise the uncertainty of the model parameters and be more in line with the actual systems [13]. In this way, the LQR method is no longer suitable for dealing with noises and uncertainties, which are strongly dependent on the system state. Thus, finding a more advanced control algorithm aiming at multiplicative noise is crucial.

Target tracking has always been a hot topic in both theoretical research and practical applications [14]. Unlike the general tracking problem, for the stochastic systems, it usually requires the system to track a distribution rather than a single value. For the Gaussian processes, the controller can be simply designed to track the mean and variance of the given distribution. For the non-Gaussian processes, there are many algorithms and literature works published under the B-spline model framework proposed by Wang. For instance, Zhou designed the observer to control the observed state to track the actual state of the system [15]. Luan added noise to the model to control the output PDF by the optimal control algorithm [16]. In most of the related studies, they only considered the static tracking targets and neglected the time-varying targets. However, in many industrial plants, in order to meet the field requirements, the tracking target can be time-varying, which increases the difficulties for controller design.

In summary, based on the aforementioned problems, the challenges of stochastic systems include, but are not limited to the following: (1) most existing stochastic control algorithms are limited to the Gaussian assumption and have poor control performance once the systems contain non-Gaussianity or nonlinearity; (2) for the published control frameworks under the B-spline approximation model proposed by Wang [8], even though they have successfully lifted the Gaussian assumption of the traditional stochastic method, they have failed to involve the uncertainties and errors of the model parameters in a stochastic dynamical manner; (3) The majority of stochastic tracking problems consider static references other than the time-varying reference, which brings problems to practical applications. To address these issues, this paper proposes a novel B-spline model-based control framework using the fully probabilistic control design (FPD) [17,18]. In this method, a linear B-spline model is applied to describe the distribution of the system output [11,16]. Besides the B-spline model, there are other approximation models that can also be used to fit the PDF curve such as the RBF model [19], the input and output ARMAX model [20], the neural network PDF model, and so on [21]. For the RBF model, the existing research shows that there are few intermediate parameters in RBF model control, but the dynamical relationship between the input and output cannot be reflected. Furthermore, the model accuracy of the RBF model is relatively poor especially when few basis functions are chosen. Although the input–output ARMAX model can reflect the dynamic relationship between the input and output to some extent, it has difficulty controlling the shape of the system when designing the controller. The neural network PDF model can achieve satisfactory control results; however, for a multi-input multi-output system, the number of basis functions will increase exponentially with the increase of the complexity of the system. Compared with the other types of approximation models mentioned above, the B-spline model is the most widely used and is easier to convert to the state-space dynamical weight model. Furthermore, compared to the other types of B-spline model such as the square root B-spline model [22], rational B-spline model [23], and rational square root B-spline model [24], the linear B-spline model is relatively simple and unique in expression. Based on the B-spline approximation principle [25], this model approximates the system output probability density function by *n* pre-selected B-spline basis functions. Due to the distribution constraints, only n−1 of the *n* weights in this model are independent of each other [26]. Therefore, the control of the system output PDF can be achieved by controlling n−1 independent weights. Thus, the control goal is transferred from altering the shape of the PDF to a desired PDF to controlling the weights of the B-spline model to a pre-selected weight set. Besides, the PDF model error is characterised as multiplicative noise [27,28], indicating that the weight error of the PDF model is involved in a dynamical manner. Considering that, the randomised controller FPD was implemented in this paper to achieve the control goal [29,30]. The FPD is evaluated by minimising the KL-divergence between the distribution of the system dynamics and the desired distribution. To better cope with multiplicative noises, the extended FPD proposed by Zhou [12] was applied in this paper. This extended FPD is strongly intuitively appealing and provides an explicit minimisation strategy that exhibits better convergence, as well as shorter response times when dealing with multiplicative noise [30].

To sum up, under the B-spline framework proposed by Wang, the purpose of this paper is to design a randomised control method so that the weight of the dynamics can track a target time-varying weight, thus realising PDF shape tracking and providing a theoretical basis for PDF tracking for non-Gaussian stochastic systems in the actual industry. The contribution of this paper can be summarised as follows:1The linear B-spline model is implemented to characterise the system PDF, thus converting the PDF shape control problem into the weight control problem;2The PDF model error is represented as multiplicative noise, indicating the stochasticity and dynamics of the weight error;3The dynamics of the weights are characterised by the stochastic state space model, thus providing full stochastic properties;4The extended FPD is applied to better cope with the multiplicative noises existing in the dynamics of the weight model. Compared with the conventional FPD [29], the extended FPD can better cope with multiplicative noises by modifying the Riccati equations. Moreover, we improved the extended FPD in [12] so that the system state can track the time-varying targets.

The rest of this paper is organised as follows. Section 2 states a description of the problem. In Section 3, the optimal control law is solved based on the performance metrics and the implementation algorithm is provided. In Section 4, the controller is applied to a numerical example and a comparison simulation with the LQR method is also included. Finally, the conclusion and future work are summarised in Section 5.

## 2. Problem Description

### 2.1. PDF Description Based on B-Spline

For the output PDF of the controlled system, if the PDF is obtained by solving partial differential equations, it is very challenging to model by first principles, thus bringing difficulties to obtaining an effective control strategy [31]. To address that, the B-spline model can be applied to fit the PDF curve by the relationship between the weights and basis functions. Assuming that the interval [a, b] is known and the output PDF γ(y) is continuous and bounded on the interval [a, b], the output PDF can be represented using *n* B-spline basis functions as follows:(1)$$\gamma \left(y\right)=\sum_{i=1}^{n}w_{i}B_{i}\left(y\right),$$
where $w_{i}$, (i=1,2,...,n) is the weight and $B_{i}\left(y\right)$ is the pre-selected *n* basis functions. There are different functions that can be chosen as basic functions such as the Gaussian function, radial basis function, and wavelet basis function. From the properties of the B-spline function, it can be used to approximate any continuous function defined on a compact set [25]. A notable advantage of the B-spline method is the exact and smooth surfacewise description of the curve [32]. Compared with the RBF basis function, when the number of basis functions is low, the B-spline basis function is more suitable for curve fitting. This will be proven later by the comparison simulation results in Section 4. Since γ(y) is a PDF defined on the interval [a, b], it should satisfy the following constraint:(2)$$\int_{a}^{b}\gamma \left(y\right)=1.$$

As Equation (Equation 2) needs to be guaranteed, it follows that only the n−1 weights are independent of each other, for which the distribution can be written in the following form: (3)$$\begin{array}{c} \gamma \left(y\right)=C_{0}\left(y\right)x+L\left(y\right), \end{array}$$(4)$$\begin{array}{c} x={\left[w_{1},w_{2},...,w_{n-1}\right]}^{T}\in R^{n-1\times 1}, \end{array}$$(5)$$\begin{array}{c} L\left(y\right)={\left(\int_{a}^{b}B_{n}\left(y\right)dy\right)}^{-1}B_{n}\left(y\right)\in R^{1\times 1}, \end{array}$$
(6)$$\begin{array}{c} C_{0}\left(y\right)={\left[\begin{array}{c} B_{1}\left(y\right)-\frac{B_{n}\left(y\right)}{\int_{a}^{b}B_{n}\left(y\right)dy}\int_{a}^{b}B_{1}\left(y\right)dy\\ B_{2}\left(y\right)-\frac{B_{n}\left(y\right)}{\int_{a}^{b}B_{n}\left(y\right)dy}\int_{a}^{b}B_{2}\left(y\right)dy\\ \vdots \\ B_{n-1}\left(y\right)-\frac{B_{n}\left(y\right)}{\int_{a}^{b}B_{n}\left(y\right)dy}\int_{a}^{b}B_{n-1}\left(y\right)dy \end{array}\right]}^{T}\in R^{1\times n-1}, \end{array}$$
where x is the weight set, $C_{0}\left(y\right)$ is the basis function vector, and L(y) is the basis function scalar. From Equations () and (Equation 6), we can see that $C_{0}\left(y\right)$ and L(y) are known once the basis functions are chosen.

According to Equations (Equation 1)–(Equation 6), we can see that, by using the B-spline model, controlling the output PDF shape can be realised by controlling n−1 independent weight vectors [33].

### 2.2. PDF Tracking Control Problem

Figure 1 demonstrates the tracking diagram of the system *B* to the system *A*. The target system *A* has an unknown structure and can monitor the output in real time, which means that for any *k* moments, the output PDF distribution $g_{k}\left(y\right)$ of the system *A* is known. System *B* is a known controlled system with control input *u* and output $\gamma_{k}\left(y,u_{k}\right)$. The goal here is to make the output distribution of System *B* track the output distribution of System *A*, where D characterises the difference between the two distributions. According to Figure 1, the system and tracking control problems mentioned above will be described in detail in the following section.

Consider the stochastic system with output PDF $\gamma_{k}\left(y\right)$, whose dynamics is described by:(7)$$\gamma_{k+1}\left(y\right)=f\left(\gamma_{k}\left(y\right),u_{k}\right),$$
where the distribution of the system output *y* is $\gamma_{k}\left(y\right)$ and $u_{k}$ is the system input.

By applying the B-spline model (Equation 3), the output PDF $\gamma_{k}\left(y\right)$ of the tracking system B is described as follows:(8)$$\gamma_{k}\left(y\right)=C_{0}\left(y\right)x_{k}+L\left(y\right),$$
where $x_{k}$ are the weights corresponding to the basis functions.

The tracking target $g_{k}\left(y\right)$ is a dynamic target PDF that varies dynamically with time in the following form:(9)$$g_{k}\left(y\right)=C_{0}\left(y\right)V_{k}+L\left(y\right),$$
where $V_{k}$ are the pre-set target weights corresponding to each basis function.

Note that the B-spline basis functions need to be selected in advance. Then, the shaping distribution problem of the output PDF $\gamma (y,u_{k})$ on the interval [a, b] can be realised by controlling the weight state $x_{k}$. Based on the B-spline model framework given in [8], the dynamics of the weight states $x_{k}$ of the B-spline model-based PDF are described as follows:(10)$$x_{k+1}=Gx_{k}+Hu_{k}+Dx_{k}E_{k},$$
where $G\in R^{n-1\times n-1}$, $H\in R^{n-1\times 1}$ are the corresponding weight system parameters and $E_{k}$ represents the state-based model randomness whose distribution is given by
(11)$$E_{k}∽\left(0,M\right),$$
where *M* is the variance of $E_{k}$.

The control flow chart is shown above in Figure 2. After pre-selecting the B-spline basis function, the time-varying target weight $V_{k}$ in Figure 2 can be obtained according to the target distribution through the B-spline principle. The system input $u_{k}$ is obtained by evaluating the target weight and the system weight through the FPD controller, which will be introduced in the next section. The weight $x_{k+1}$ is then updated by B-spline principle modelling and input $u_{k}$. According to the relationship between the weight and the basis function, the output distribution is then obtained. In addition, it should be noted that the model error part $E_{k}$ is represented by multiplicative noise, and D is the weight matrix with the appropriate dimension. Finally, the weights are iteratively updated according to the model to achieve output distribution control.

Following this sequence, our control goal was to design a randomised controller so that the weight $x_{k}$ can follow the target weight $V_{k}$. The PDF shape-tracking control problem is transformed into a weight tracking control problem. The controller design will be introduced in the next section.

## 3. Control Algorithm

In this section, the FPD control algorithm is introduced to achieve the tracking goals for the weight state. The main reason we chose FPD is that we considered the multiplicative noise in the weight dynamical system. Moreover, FPD not only fully respects the stochastic nature of the system, but also provides a very detailed implementation procedure. The details will be given as follows.

### 3.1. General Control Solution of FPD

In FPD, the difference between the actual distribution f(D) of the system and the target distribution $f^{I}\left(D\right)$ is measured by the Kullback–Leibler divergence (KLD) [34], where $D=(x_{k},u_{k})$ represents the observation data. The closer the distance between the output distribution and the target distribution, the greater the degree of similarity and, conversely, the smaller the degree of similarity [35]. The formula for the KLD is given by
(12)$$D(f||f^{I})=\int f\left(D\right)ln\frac{f(D)}{f^{I}\left(D\right)}dD,$$
where D is the relative entropy, directional divergence, or KLD and D has the key property of non-negativity, i.e., $D(f||f^{I})\geq 0$.

Assuming that $u_{k}$ is the input to the system and $k\in k_{0},k_{1},...,k_{n}$, all states $x_{k}$ of the system are assumed to be measurable; the joint PDF of the closed-loop system from moment $k_{0}$ to moment $k_{n}$ is:(13)$$f\left(D\right)=\prod_{k=k_{0}}^{k_{n}}s\left(x_{k}|x_{k-1},u_{k-1}\right)c\left(u_{k-1}|x_{k-1}\right),$$
where $s(x_{k}|x_{k-1},u_{k-1})$ denotes the distribution of the system dynamics at moment *k* and $c(u_{k-1}|x_{k-1})$ denotes the distribution of the controller at moment *k*. Similar to the joint PDF system, the target PDF $f^{I}\left(D\right)$ is given by
(14)$$f^{I}\left(D\right)=\prod_{k=k_{0}}^{k_{n}}s^{I}\left(x_{k}|x_{k-1},u_{k-1}\right)c^{I}\left(u_{k-1}|x_{k-1}\right),$$
where $s^{I}\left(x_{k}|x_{k-1},u_{k-1}\right)$ denotes the desired distribution at moment *k*, while $c^{I}\left(u_{k-1}|x_{k-1}\right)$ denotes the desired distribution of the controller at moment *k*.

The control strategy is to find an optimal randomised controller to bring the distribution of the system dynamics as close as possible to a target distribution, thus minimising the KLD given in Equation (Equation 12). The following performance indicators can be established according to Equations (Equation 12)–(Equation 14):(15)$$\begin{array}{c} -ln\left(\gamma \left(x_{k-1}\right)\right)=\min_{\left\{c{\left(u_{k-1}|x_{k-1}\right)}_{\tau >k}^{k_{n}}\right\}}\sum_{\tau =k}^{k_{n}}\int f\left(D_{\tau }|x_{k-1}\right)\times \\ ln\left(\frac{s\left(x_{\tau }|x_{\tau -1},u_{\tau -1}\right)c\left(u_{\tau -1}|x_{\tau -1}\right)}{s^{I}\left(x_{\tau }|x_{\tau -1},u_{\tau -1}\right)c^{I}\left(u_{\tau -1}|x_{\tau -1}\right)}\right)d\left(D_{\tau }\right), \end{array}$$
where, for any $\tau \in k_{0},k_{1},...,k_{n}$, the recursive form of Equation (Equation 15) can be obtained according to the dynamic programming method as follows:(16)$$\begin{array}{cc} -ln(\gamma \left(x_{k-1}\right)) & =\min_{c(u_{k-1}|x_{k-1})}\int s\left(x_{k}|x_{k-1},u_{k-1}\right)c\left(u_{k-1}|x_{k-1}\right)\\ \; & \times \left[ln(\frac{s\left(x_{k}|x_{k-1},u_{k-1}\right)c\left(u_{k-1}|x_{k-1}\right)}{s^{I}\left(x_{k}|x_{k-1},u_{k-1}\right)c^{I}\left(u_{k-1}|x_{k-1}\right)}-ln\left(\gamma \left(x_{k}\right)\right)\right]d\left(x_{k},u_{k-1}\right). \end{array}$$

Thus, the optimal control law $c^{*}\left(u_{k-1}|x_{k-1}\right)$ that minimises the performance index (Equation 16) can then be evaluated as follows based on FPD:(17)$$c^{*}\left(u_{k-1}|x_{k-1}\right)=\frac{c^{I}\left(u_{k-1}|x_{k-1}\right)exp\left[-\beta_{1}\left(u_{k-1},x_{k-1}\right)-\beta_{2}\left(u_{k-1},x_{k-1}\right)\right]}{\gamma (x_{k-1})},$$
where $\gamma \left(x_{k-1}\right)=\int c^{I}\left(u_{k-1}|x_{k-1}\right)exp\left[-\beta_{1}\left(u_{k-1},x_{k-1}\right)-\beta_{2}\left(u_{k-1},x_{k-1}\right)\right]du_{k-1}$,

$\beta_{1}\left(u_{k-1},x_{k-1}\right)=\int s\left(x_{k}|x_{k-1},u_{k-1}\right)\left[ln\frac{s(x_{k}|x_{k-1},u_{k-1})}{s^{I}\left(x_{k}|x_{k-1},u_{k-1}\right)}\right]dx_{k}$,

$\beta_{2}\left(u_{k-1},x_{k-1}\right)=-\int s\left(x_{k}|x_{k-1},u_{k-1}\right)ln\left(\gamma \left(x_{k}\right)\right)dx_{k}$.

The proof of Equation (Equation 17) can be found in [12].

### 3.2. FPD Control Solution for the Weight Dynamic System

To apply the FPD, all the variables will be characterised as stochastic distributions. Based on Equation (Equation 10), the weight dynamic distribution is given as follows with $\mu_{k}$ as the mean and $Z_{k}$ as the covariance:(18)$$s\left(x_{k}|x_{k-1},u_{k-1}\right)∽N\left(\mu_{k},Z_{k}\right),$$
where
$$\begin{array}{cc} \mu_{k} & =Gx_{k-1}+Hu_{k-1},\\ Z_{k} & =cov(x_{k}|x_{k-1},u_{k-1})\\ \; & =E\left\{\left(x_{k}-\mu_{k}\right){\left(x_{k}-\mu_{k}\right)}^{T}\right\}\\ \; & =E\left\{Dx_{k-1}E_{k-1}^{T}x_{k-1}^{T}D^{T}\right\}\\ \; & =Dx_{k-1}Mx_{k-1}^{T}D^{T}. \end{array}$$

The ideal weight distribution is assumed to be
(19)$$s^{I}\left(x_{k}|x_{k-1},u_{k-1}\right)∽N\left(V_{k},R_{k}\right),$$
where $V_{k}$ is the target weight of the system distribution at instant *k* and $R_{k}$ is its ideal covariance. Note that $V_{k}$ is dynamic and time-varying, so as to better simulate the real situation.

The distribution of the ideal controller can also be formulated as
(20)$$c^{I}\left(u_{k-1}|x_{k-1}\right)∽N\left(\theta_{k-1},\Gamma \right),$$
where Γ is the ideal covariance of the controller and $\theta_{k-1}$ is the ideal mean of the optimal control signal at each moment of the controller, which can be evaluated as follows:(21)$$\begin{array}{cc} \theta_{k-1} & ={\left(H^{T}H\right)}^{-1}H^{T}\left(V_{k}-GV_{k-1}\right). \end{array}$$

Therefore, given the system output distribution (Equation 18) and the target distribution (Equation 19) and (Equation 20), following the general PFD scheme Equation (Equation 17), the optimal distribution of the tracking controller is given by the following theorem.

**Theorem** 1.
*Under the PDF (Equation 18) describing the dynamic weights of the system, the optimal controller distribution for minimising the performance index function (Equation 17) is given as follows:*

(22)
$$c^{*}\left(u_{k-1}|x_{k-1}\right)∽N\left({\bar{u}}_{k-1},\Gamma_{k-1}\right),$$


*where*

(23)
$$\begin{array}{cc} {\bar{u}}_{k-1} & =-K_{k-1}x_{k-1}+d_{k-1},\\ K_{k-1} & =\Gamma_{k-1}H^{T}Q_{k}G,\\ Q_{k} & =R_{k}^{-1}+S_{k},\\ \Gamma_{k-1} & ={\left(\Gamma^{-1}+H^{T}Q_{k}H\right)}^{-1},\\ d_{k-1} & =\Gamma_{k-1}\left[\Gamma^{-1}\theta_{k-1}+H^{T}R_{k}^{-1}V_{k}-0.5H^{T}P_{k}^{T}\right], \end{array}$$


*and we establish the following performance indicator function:*

(24)
$$-ln\left(\gamma \left(x_{k-1}\right)\right)=0.5x_{k-1}^{T}S_{k-1}x_{k-1}+0.5P_{k-1}x_{k-1}+0.5q_{k-1},$$


*where*

(25)
$$\begin{array}{cc} S_{k}= & -G^{T}Q_{k}H\Gamma_{k-1}H^{T}Q_{k}^{T}G+G^{T}Q_{k}G+Dx_{k-1}Mx_{k-1}^{T}D^{T},\\ P_{k-1}= & \left(P_{k}-2V_{k}^{T}R_{k}^{-1}\right)G+2\theta_{k-1}^{T}\Gamma^{-1}\Gamma_{k-1}H^{T}Q_{k}G+2V_{k}^{T}R_{k}^{-1}H\Gamma_{k-1}H^{T}Q_{k}G\\ \; & -P_{k}H\Gamma_{k-1}H^{T}Q_{k}G, \end{array}$$

*where Equation (Equation 24) is the Riccati expression of the performance index, $S_{k}$ is the quadratic Riccati equation, $P_{k}$ is the linear term of the Riccati equation, and $q_{k-1}$ stands for some normal numbers that do not contribute to the controller, thus omitted here. In addition, ${\bar{u}}_{k-1}$ is the mean of the optimal controller, $\Gamma_{k-1}$ stands for the covariance, and $K_{k-1}$ is the controller feedback gain. Moreover, $d_{k-1}$ is a linear shift caused by the considered tracking control problem.*


**Proof.** The proof of this theorem can be evaluated following the same procedure in our previous publication [12], thus omitted here.    □

**Remark** 1.
*The FPD algorithm given by Equation (Equation 17) is the general solution to the fully probabilistic control design irrespective of the type of distribution describing the system dynamics or whether the system is linear or nonlinear. It can be derived numerically or analytically. The specific solution of the FPD given by Theorem 1 is the analytical solution for this specific case.*


The proposed control framework for the B-spline modelled PDF shaping using the FPD method is summarised by Algorithm 1.
**Algorithm 1:** Tracking control framework with output probability density function.
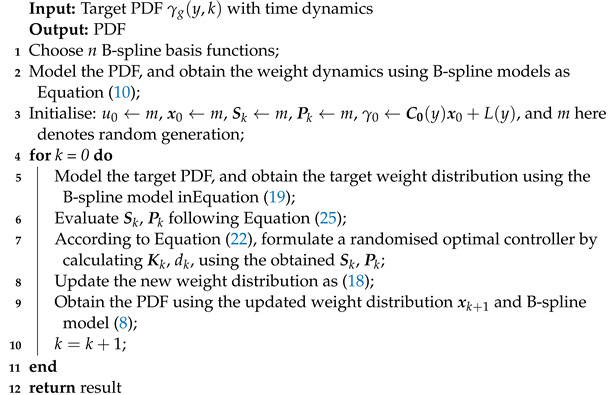


## 4. Simulation Result

In this section, the proposed framework is tested on a numerical example to illuminate the effectiveness of the proposed control method. Moreover, to better demonstrate the advantages of the proposed framework, two comparison simulations are also included.

For the discrete system, the following B-spline basis functions were selected to approximate the linear model.
(26)$$\begin{array}{cc} B_{1}\left(y\right) & =0.5\left(y^{2}+6y+9\right)I_{1}+\left(-y^{2}-3y-1.5\right)I_{2}+0.5y^{2}I_{3},\\ B_{2}\left(y\right) & =0.5\left(y^{2}+4y+4\right)I_{2}+\left(-y^{2}-y+0.5\right)I_{2}+0.5\left(y^{2}-2y+1\right)I_{4},\\ B_{3}\left(y\right) & =0.5\left(y^{2}+2y+1\right)I_{3}+\left(-y^{2}+y+0.5\right)I_{4}+0.5\left(y^{2}-4y+4\right)I_{5},\\ B_{4}\left(y\right) & =0.5y^{2}I_{4}+\left(-y^{2}+3y-1.5\right)I_{5}+0.5\left(y^{2}-6y+9\right)I_{6}, \end{array}$$
where $I_{i}=\left\{\begin{array}{cc} 1 & y\in [i-4,i-3]\\ 0 & others \end{array}\right.$, i=1,...,6.

Firstly, the comparison simulation of the approximation between the B-spline model and the RBF model with the following Gaussian basis function is presented to show the advantages of the B-spline model.
(27)$$\begin{array}{c} h_{j}\left(y\right)=exp\left(-\frac{{\left(y-c_{j}\right)}^{2}}{b_{j}^{2}}\right)\;i=1,...,n_{j}, \end{array}$$
where *j* is the order of nodes, $n_{j}$ is the total number of nodes of the RBF basis function, $c_{j}$ and $b_{j}$ are the centre value and width, and variable y∈[a,b]. The same as the B-spline model, four basis functions of the RBF model were chosen to fit the target PDF curve. The simulation result is shown in Figure 3 and Figure 4. Figure 3 shows the fitting effect of the B-spline basis function and RBF basis function on the curve, where the red real line is the fitting curve of the B-spline model, the green real line is the fitting curve of the RBF model, and the black dotted line is the target curve. Figure 4 demonstrates the fitting errors of the two models, where the red line is the fitting error of the B-spline model and the green solid line is the fitting error of the RBF model. From Figure 3, it can be seen that, for the same number of basis functions, the B-spline model has a better fitting performance than the RBF model. It can also be proven by Figure 4, which shows clearly that the B-spline model has a much smaller fitting error than the RBF model. Therefore, the result proves that the B-spline model is more suitable for PDF fitting when the number of basis functions is low. As a state space model, the weight dynamic system from the approximation model generally has a lower number of basis functions, which makes the B-spline model more suitable in such cases.

To verify the control effectiveness of the proposed control algorithm, the following example, which uses the same four basis functions (Equation 26), is given as below. Due to the existence of the constraint condition (Equation 2), only three corresponding weights will be required out of four B-spline basis functions. The fourth weight is linearly related to the first three weights so that the model order is reduced to three. The coefficient matrix of the system model is
(28)$$\begin{array}{c} x_{k+1}=Gx_{k}+Hu_{k}+Dx_{k}E_{k}, \end{array}$$
with $G=\left[\begin{array}{ccc} 0.555 & -0.098 & -0.041\\ -0.1 & -0.734 & 0.181\\ -0.292 & 0.02 & 0.291 \end{array}\right]$, $H=\left[\begin{array}{c} 0.275\\ 0.302\\ 0.302 \end{array}\right]$, $D=\left[\begin{array}{ccc} 0.41 & 1.66 & 0.51\\ -0.11 & 0.215 & 0.16\\ 0.31 & 0.02 & -1.005 \end{array}\right]$, where G is the state weight matrix, H is the control matrix, D is the random weighting matrix of the noise term, and $E_{k}$ is the Gaussian noise. The target weight is given by $V_{g}={\left[\begin{array}{ccc} 0.4229 & 0.1217 & 0.1487 \end{array}\right]}^{T}$ for the first 100 steps and $V_{g}={\left[\begin{array}{ccc} 0.5908 & 0.1701 & 0.2077 \end{array}\right]}^{T}$ for the remaining steps. In addition, the system initial weight state is $x_{0}={\left[\begin{array}{ccc} 0.2673 & 0.0969 & 0.1897 \end{array}\right]}^{T}$; the covariance of the noise is subjected to the distribution given by $E_{k}∽N\left(0,0.004\right)$; the ideal state covariance R was chosen as $R=diag\left[\begin{array}{ccc} 0.4 & 0.004 & 10 \end{array}\right]$.

In order to verify the tracking performance of the algorithm proposed in this paper, the results were compared with the conventional LQR algorithm. The following simulation results can be obtained following Algorithm 1. The state feedback gain matrix $K=\left[\begin{array}{ccc} -0.2544 & -2.2287 & 0.535 \end{array}\right]$ is calculated according to Equation (Equation 23). The state feedback gain matrix $K_{L}=\left[\begin{array}{ccc} 0.2814 & -0.9455 & 0.3609 \end{array}\right]$ is calculated according to LQR algorithm. Figure 5 shows the four weights of the FPD algorithm and LQR algorithm and their target reference curves, respectively. More specifically, in Figure 5, the red line is the weight-tracking curve under the FPD algorithm, while the blue line represents the weight-tracking curve under the LQR algorithm. Note that the weights $V_{1}$, $V_{2}$, and $V_{3}$ are the controlled states, while the weight $V_{4}$ has a linear relationship with the other three weights, which is not included as the controlled state due to the existence of the constraints. Therefore, the weight $V_{4}$ is represented by a dotted line. From Figure 5, it can be seen that, even with the effect of the multiplicative noises, both the FPD algorithm and the LQR algorithm can successfully track the given references and manage to keep tracking the new reference values when the target changes at the 100th step. Compared with the LQR algorithm, we can see that the FPD algorithm shows a better tracking effect and smaller tracking error under the effect of multiplicative noise. Figure 6 indicates the system control inputs, where the red line is the control input curve under the FPD algorithm and the blue line is the control input curve under the LQR algorithm. The results of Figure 6 show that, due to the change in the shape of the target, the input of the control target will also change synchronously. Compared with the LQR algorithm, the input of the FPD algorithm contains less randomness under multiplicative noise. Figure 7 shows the iterative process of the two algorithms controlling the output PDF curve. Compared with the results of the system under the action of the two algorithms, the FPD algorithm has a smaller error and a smoother tracking process from the initial curve to the target curve. When the target curve changes, the system control output curve will still track the new target. Therefore, it can be concluded that, compared with the LQR algorithm, the extended FPD algorithm can show a better tracking effect in the dynamic random B-spline function-based weight dynamical system with multiplicative noise and time-varying target.

## 5. Conclusions

This paper investigated the output distribution shape-tracking problem for a class of stochastic distribution systems under the B-spline model framework. The linear B-spline model was implemented to fit the PDF curve and simplified the PDF shaping problem to a dynamic weight-altering problem. At the same time, the randomness and the model error of the dynamic weight system were characterised as state-dependent noises, which more realistically simulate the actual complex system. In addition, the target distribution shape of the system was changed from a fixed shape to a time-varying shape, which makes the control goal more realistic and convenient to operate in the actual working process. The extended FPD algorithm was then implemented to achieve the time-varying tracking goal under the effect of the multiplicative noises. Moreover, the implementation procedure was provided step by step. Finally, the simulation results were obtained following the implementation procedure. Meanwhile, to make the experimental results more convincing, the conventional LQR algorithm was also added for comparison. As a result, the simulation showed that the proposed control framework can make the dynamic weights track their time-varying targets under the effect of multiplicative noises. Compared with the LQR algorithm, the proposed extended FPD algorithm has smaller tracking errors and stronger robustness to the multiplicative noises in the model. In addition, in order to better illustrate the suitability of the B-spline model selected in this paper, the RBF model was also added for comparison. By analysing the simulation results, it was concluded that the B-spline model has a smaller fitting error than the RBF model when the number of basis functions is small.

Overall, the method proposed in this paper can effectively solve the time-varying distribution shape-tracking problem for a class of stochastic systems. In practice, the application of the B-spline framework can productively model the non-Gaussian PDF, where most of the practical systems contain non-Gaussian variables. Moreover, using stochastic distributions to characterise the dynamics of weights can fully represent the stochastic properties. In addition, the proposed controller structure is simple and easy to follow for practical application. Thus, the proposed control method can further promote the application of PDF control. In future work, we will consider the application of the proposed framework to real-world systems.

## Figures and Tables

**Figure 1 entropy-25-00186-f001:**
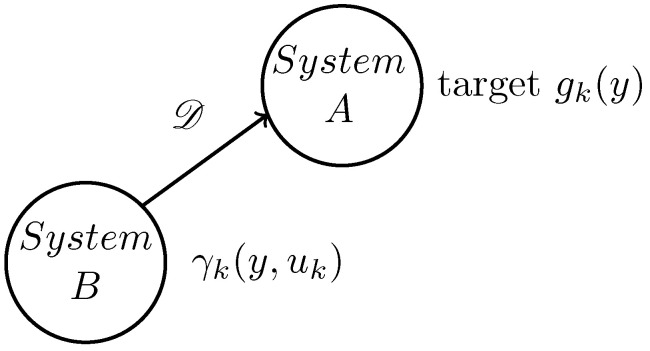
Diagram of the tracking system.

**Figure 2 entropy-25-00186-f002:**
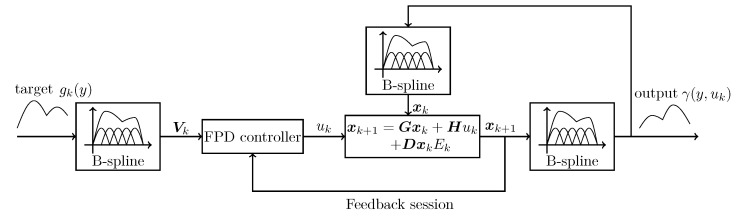
System control structure diagram.

**Figure 3 entropy-25-00186-f003:**
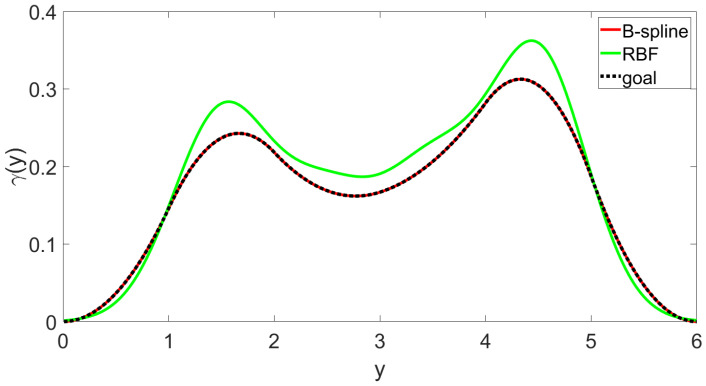
B-spline and RBF basis function fitting curve.

**Figure 4 entropy-25-00186-f004:**
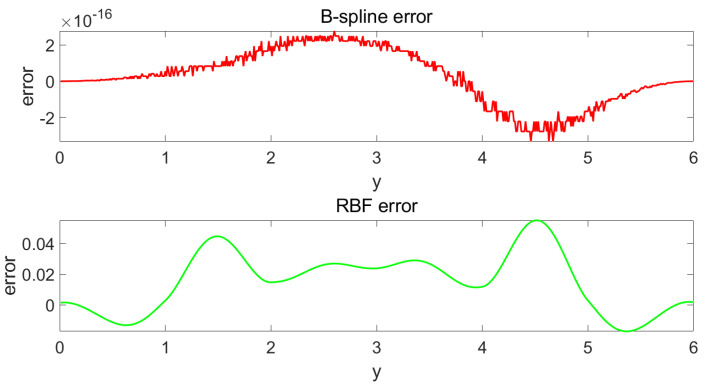
Error of fitting.

**Figure 5 entropy-25-00186-f005:**
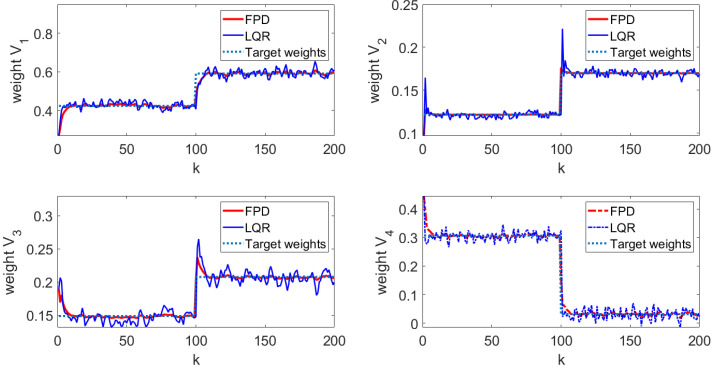
Weight-tracking curve.

**Figure 6 entropy-25-00186-f006:**
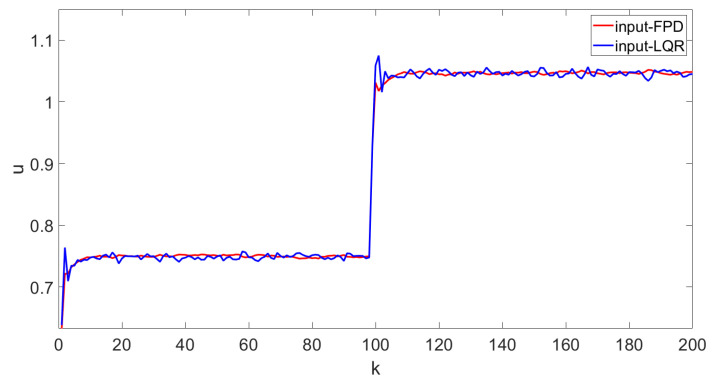
Control input curve.

**Figure 7 entropy-25-00186-f007:**
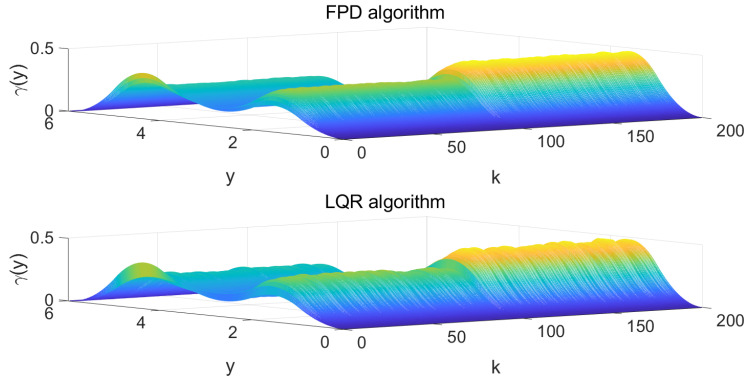
System output PDF of 3D drawings.

## Data Availability

Not applicable.

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
