# Peer review of "Tracking Control for Output Probability Density Function of Stochastic Systems Using FPD Method"

_entropy, 2023, doi:10.3390/e25020186_

Round 1
Reviewer 1 Report
This paper proposes a novel stochastic control framework so that the output PDF can track a given time-varying PDF by using B-spline model approximation. In my point of view, the motivation of the practical use of the theoretic results obtained should be clearly addressed. The best way to show this is by practical example or explanations. This work can not be accepted with out any comparison with available schemes such as same method with out using B-spline model approximation. It should be clearly shown that why to use B-spline approximation is appropriate. Is it good when the model faces with uncertainty or ... ?
Reviewer 2 Report
The work is relevant and intended to describe stochastic processes, but has many shortcomings:
1. The introduction is general. It is necessary to expand the introduction and describe the application of stochastic processes and their models in practice, specifically illustrating with examples. The goal is not specific, it is not clear what the work is intended to achieve and what its practical value is.
2. When describing the algorithm, it is recommended to also describe what methods were used to solve the resulting systems of equations (analytical, numerical, etc.).
Figures 3, 4 and 5 are not described at all. Their purpose in this work is not clear at all. By the way values of the on the axles are small and invisible.
3. Conclusions are abstract and not specific. The authors state that they examined the shape of the output trace distribution, this is not a conclusion. The conclusions should be specified by emphasizing the essential results and regularities obtained. By the way has such statements as e.g. "Finally, the simulation results indicated that the proposed 179
control framework shows good performance in dealing with PDF tracking problems"......such statements are vague and say nothing.
The article is currently not suitable for publication.
Round 2
Reviewer 1 Report
Authors have addressed all the comments raised and have made changes accordingly.
Author Response
Thanks very much for your approval.
Reviewer 2 Report
I am grateful that the authors took my comments into account and substantially corrected this article. The new version of the article is significantly better than the old one, but there are still a few additions to be made:
1. At the end of the introduction, it is proposed to specifically and clearly (concisely) define the purpose of the work (now it is difficult to understand what the work is intended to achieve).
2. In the conclusions, it is recommended to describe how the results or regularities obtained by the authors really help to solve various modeling problems using the density function in practice.
Round 3
Reviewer 2 Report
I thank the authors for taking into account my recommendations. I have no further comments, I think the article is worth publishing.